# Oxidative Stress, Telomere Shortening, and Apoptosis Associated to Sarcopenia and Frailty in Patients with Multimorbidity

**DOI:** 10.3390/jcm9082669

**Published:** 2020-08-18

**Authors:** Máximo Bernabeu-Wittel, Raquel Gómez-Díaz, Álvaro González-Molina, Sofía Vidal-Serrano, Jesús Díez-Manglano, Fernando Salgado, María Soto-Martín, Manuel Ollero-Baturone

**Affiliations:** 1Internal Medicine Department, Hospital Universitario Virgen del Rocío, 41013 Sevilla, Spain; aglezmolina@gmail.com (Á.G.-M.); m.ollero.baturone@gmail.com (M.O.-B.); 2General and Multiple Use Laboratory, Instituto de Biomedicina de Sevilla, 41013 Sevilla, Spain; rgomez-ibis@us.es; 3Internal Medicine Department, Hospital San Juan de Dios del Aljarafe, 41930 Sevilla, Spain; sofiavidalserrano@gmail.com; 4Internal Medicine Department, Hospital Royo Villanova, 50015 Zaragoza, Spain; jdiez@aragon.es; 5Internal Medicine Department, Hospital Regional, 29010 Málaga, Spain; fersalord@gmail.com; 6Internal Medicine Department, Hospital Juan Ramón Jiménez, 21005 Huelva, Spain; msoto@hotmail.com

**Keywords:** multimorbidity, polypathological patients, sarcopenia, frailty, oxidative stress, telomere length, apoptosis

## Abstract

Background: The presence of oxidative stress, telomere shortening, and apoptosis in polypathological patients (PP) with sarcopenia and frailty remains unknown. Methods: Multicentric prospective observational study in order to assess oxidative stress markers (catalase, glutathione reductase (GR), total antioxidant capacity to reactive oxygen species (TAC-ROS), and superoxide dismutase (SOD)), absolute telomere length (aTL), and apoptosis (DNA fragmentation) in peripheral blood samples of a hospital-based population of PP. Associations of these biomarkers to sarcopenia, frailty, functional status, and 12-month mortality were analyzed. Results: Of the 444 recruited patients, 97 (21.8%), 278 (62.6%), and 80 (18%) were sarcopenic, frail, or both, respectively. Oxidative stress markers (lower TAC-ROS and higher SOD) were significantly enhanced and aTL significantly shortened in patients with sarcopenia, frailty or both syndromes. No evidence of apoptosis was detected in blood leukocytes of any of the patients. Both oxidative stress markers (GR, *p* = 0.04) and telomere shortening (*p* = 0.001) were associated to death risk and to less survival days. Conclusions: Oxidative stress markers and telomere length were enhanced and shortened, respectively, in blood samples of polypathological patients with sarcopenia and/or frailty. Both were associated to decreased survival. They could be useful in the clinical practice to assess vulnerable populations with multimorbidity and of potential interest as therapeutic targets.

## 1. Introduction

As a result of populations’ aging throughout the world, the prevalence of chronic conditions has drastically increased; these coexist frequently in the same patient, conditioning deleterious relationships, faster clinical and functional deterioration, poorer quality of life, and higher mortality. Taking multimorbidity seriously is of nuclear importance for the sustainability of all healthcare systems [1,2,3]. Multimorbidity is narrowly correlated to aging. As a matter of fact, there is a direct and strong correlation between the development of different chronic conditions and longevity. The easy explanation of this correlation is the longer exposure to different risk factors (environmental agents, unhealthy lifestyles, inherited risk factors, overuse deterioration), which impact in the development of diseases and failures of multiple organs and systems [3,4].

Sarcopenia and frailty are two major geriatric syndromes closely related to the aging process [5,6]. The development of one or both of them is linked to progressive functional disability, loss of quality of life and death. Their prevalence in elderly populations approximates 10% and 15%, respectively; however, in the presence of chronic conditions and multimorbidity, these prevalences can raise to 20% and 60%, respectively [7]. Both syndromes are narrowly interrelated; as a matter of fact, they have currently an identical therapeutic approach based on physical activity and optimal nutrition. In a recent study, both sarcopenia and frailty were present in the same patient in 18% of the studied cases; that is to say that most sarcopenic patients were frail, and about one third of frail patients were sarcopenic [7]. Nevertheless, these percentages are different in other studies, probably due to sample selection criteria [8].

Both syndromes have commonalities sharing a nuclear issue, which is the physical function impairment, usually assessed by different tools like walking speed and hand grip strength. Such impairment may be responsible for the concurrent existence of a disability in both phenotypes, but they also express differences; as a matter of fact, sarcopenia rather tends to assume the lineaments of cachexia and “muscle wasting”, whereas frailty status is largely dominated by a low physical performance, homeostasis disruption to stressors, and disabling condition.

The deep and intimal relation between sarcopenia and frailty probably reflects that they share similar or identical pathophysiological routes and molecular mechanisms. In this field, many metabolic imbalances and other molecular factors have been studied and correlated in some ways to both geriatric syndromes. Sarcopenia has been associated to genetic expression of apoptosis and muscular autophagy, muscle androgenic and vitamin D receptors, chronic inflammation, oxidative stress, and telomere shortening [9,10,11]. On the other hand, frailty has been associated to inflammation pathways (demonstrated in the case of C-reactive protein, interleukin-1β, the IL-1 receptor antagonist, IL-18, and tumor necrosis factor alpha), unspecific immunological alterations linked to immunosenescence (mainly thymus involution and the corresponding decrease of T and B lymphocyte precursors and the reduction in the proliferative capacity of the T and B lymphocytes), and oxidative stress [12,13,14].

From these data, the narrow relation between frailty and sarcopenia can be extracted. This is more so in patients with multimorbidity, in which aging and chronic conditions may trigger more oxidative stress, telomere shortening, and apoptosis. In these patients, sarcopenia and frailty could be the results of a multisite “rusting” produced by chronic inflammation processes and their consequent imbalance between the production of reactive oxygen species (ROS) and cellular antioxidant defenses, present in chronic neurological, pulmonary, and cardiovascular diseases, along with atherosclerosis, diabetes, obesity, and arthritis. Nevertheless, the role and weight of any of these molecular alterations in sarcopenic and/or frail populations with multimorbidity remain unknown.

For all these reasons, we have explored the main oxidative stress markers, telomere length, and apoptosis parameters in a hospital-based multicenter cohort of multimorbidity patients. We hypothesized that all these biological markers have a deep impact and association to sarcopenia and frailty.

## 2. Patients and Methods

### 2.1. Development of the Study

This was a prospective observational, multi-institutional (6 centers) study carried out by researchers from the Polypathological Patient and Advanced Age Study Group of the Spanish Society of Internal Medicine (all participant centers are listed on the PROTEO Researchers list). The study was approved by the ethics committee of all participant centers. The study inclusion period ranged from January 2012 to March 2016.

All patients treated in the Internal Medicine and Geriatric areas who accomplished inclusion criteria (≥18 years old and fulfilling criteria of polypathological patient (PP)) were included, after providing their written informed consent. The patient’s sample was collected by performing prevalence surveys every 14 days during the study period. A total of 155 surveys were performed (29 ± 19 surveys per hospital).

After receiving informed consent, a complete set of demographical, socio-familial, clinical, functional, biological, and pharmacological data were collected from all included patients.

Sarcopenia was defined following EWGSOP criteria [15]. This was established by the presence of a gait speed ≤ 0.8 m/seg, plus a skeletal muscle mass <6.76 Kg/m^2^ in women, and <10.76 Kg/m^2^ in men (for those patients able to walk) or a hand grip strength lower than 50 percentile of his/her age group and gender, and a skeletal muscle mass <6.76 Kg/m^2^ in women and <10.76 Kg/m^2^ in men (for those patents unable to walk). Frailty was defined when fulfilling 3 or more of Fried’s criteria (slowness, weakness, weight loss, exhaustion, and low physical activity) [16].

All patients were clinically followed during a 12-month period in order to assess mortality, as previously described [7]. Time survival was assessed, and in case of death, chronology of the demise was incorporated. Therefore, we looked at mortality as a time-dependent outcome. For the dichotomous outcome, subjects were categorized depending on whether or not they survived 12 months from their initial interview date. For the continuous outcome, survival time was defined as the number of days between the baseline interview and the date of death. All these data were collected by clinicians in charge who were active members of the investigation team.

Ethics Committee Approval: The present study has been approved by the ethics committee of all participant centers (ethical approval code: CEI2012/PI242). Ethical Guidelines for Authorship and Publishing: The authors certify that they comply with the ethical guidelines for publishing in the Journal Clinical Medicine.

### 2.2. Biological Parameters Determination

We determined blood or plasma biological parameters of all included patients, including oxidative stress markers, apoptosis expression, and telomere length.

Oxidative stress markers: We determined activity/levels of catalase, glutathione reductase (GR), total antioxidant capacity to reactive oxygen species (TAC-ROS), and superoxide dismutase (SOD). Colorimetric studies were performed using a monochromator-based UV–VIS spectrophotometer (Multiskan^®^ GO; Thermo Fisher Scientific Corporation, Carlsbad, CA, USA).

Catalase activity (nmol/min/mL) was measured in patients’ plasma using the colorimetrical procedure provided by Cayman’s Catalase Assay Kit, Item No. 707002 (Cayman Chemical, Ann Arbor, MI, USA). The method is based on the reaction of the enzyme wit methanol in the presence of an optimal concentration of H_2_O_2_. The formaldehyde produced is measured colorimetrically with Purpald as the chromogen. Purpald specifically forms a bicyclic heterocycle with aldehydes, which upon oxidation changes from colorless to purple color [17,18]. 

Glutathione reductase activity (U/mL; 1 Unit = the amount of enzyme that will cause the oxidation of 1.0 nmol of NADP to NADP+ per minute at 25 °C) was analyzed in patients’ plasma by measuring the rate of NADPH oxidation, using for this purpose the Cayman’s Glutathione reductase Assay Kit, Item No. 703202 (Cayman Chemical, Ann Arbor, MI, USA). The oxidation of NADPH is accompanied by a decrease in absorbance at 340 nm and is directly proportional to the GR activity in the sample [18,19].

Total antioxidant capacity to reactive oxygen species (mM Trolox equivalents) was analyzed measuring the ability of patients’ plasma antioxidants to inhibit the oxidation of ABTS^®^ (2,2′-Azino-di-3-ethylbenzthiazoline sulphonatel) to ABTS^®^+ by metmyoglobin. For this purpose, the Cayman’s Antioxidant Assay Kit, Item No. 709001 (Cayman Chemical, Ann Arbor, MI, USA) was used. The antioxidants cause suppression of the absorbance at 750 nm or 405 nm to a degree that is proportional to their concentration. This capacity of the antioxidants is compared to that of Trolox, a water-soluble tocopherol analogue, and is quantified as millimolar Trolox equivalents [20,21].

Superoxide dismutase activity (U/mL) was measured in patients’ plasma using the colorimetrical absorbance procedure provided by Cayman’s Superoxide Dismutase Assay Kit, Item No. 706002 (Cayman Chemical, Ann Arbor, MI, USA). The method utilizes a tetrazolium salt for detection of superoxide radicals generated by xanthine oxidase and hypoxanthine. One unit of SOD is defined as the amount of enzyme needed to exhibit 50% dismutation of the superoxide radical. This assay measures all types of SOD (Cu/Zn-SOD, Mn-SOD, and Fe-SOD) [18].

Telomere length: We assessed telomere length following the procedure described by O’Callaghan and Fenech, in which the absolute telomere length (aTL) was measured [22]. For this purpose, we used Telomere standard Human/rodent (teloF and teloR) as primers for telomere length (TL) analysis and 36B4 standard human primers for single copy gene (SCG) determinations. All these were supplied by TaqMan™ Array Human Telomere Extension by Telomerase (Thermofisher Scientific, Waltham, MA, USA).

First standard curves were constructed for both experiments (TL and SCG). Then, all patients’ samples were analyzed, and aTL was calculated dividing the absolute result of TL by the result of SCG. This result was again divided by 92 (each somatic human cell has 46 chromosomes, and each chromosome has 2 telomeres) in order to obtain the mean aTL per single telomere [22].

Apoptosis: In order to detect apoptosis, we evaluated possible DNA fragmentation in patients’ leucocytes. For this purpose, we performed a DNA conventional constant field gel electrophoresis loading in a 0.8% agarose gel panel a total or 300 ng from a normalized purified DNA mixture with a DNA concentration of 30 ng/uL. DNA was purified by means of standard techniques already established [22]. When apoptosis is present, the result is fragmentation of DNA into multiples of 180 base-pair lengths; a characteristic “ladder” effect is obtained when these fragments are resolved in the agarose gel electrophoresis [23].

Statistical analysis: The dichotomous variables were described as whole numbers and percentages, and the continuous variables as mean and standard deviation (or median and interquartile rank (IQR) in those with no criteria of normal distribution). The distribution of all variables was analyzed with the Kolmorogov–Smirnov test. Possible biological parameters associated to the presence of sarcopenia and death were investigated performing the Student’s *t* for normally distributed quantitative variables, and Mann–Whitney *U* test.

Finally, we also evaluated the association of these biological parameters with functional status (by means of basal Barthel index), death risk (by means of PROFUND index), and survival (considering death as a time-dependent variable), using linear regression models. Statistical significance was considered when obtained *p* values were ≤0.05. Statistics were performed using the SPSS 22.0 software (IBM, Armonk, NY, USA).

## 3. Results

We included 444 patients with a mean age of 77.3 ± 8.4 years. Fifty-five percent were male. The main clinical features and biological parameters of the recruited patients are detailed in Table 1. Sarcopenia was present in 97 (21.8%), frailty in 278 (62.6%), and the remaining 69 (15.6%) were robust. Eighty patients (18% of the whole cohort) out of those with sarcopenia or frailty had simultaneously both phenotypes.

And combined sarcopenia and frailty were present in 80 (18%) patients. Mortality in the 12-months follow-up period was 40% (*N* = 178). A detailed clinical description of the included patients has already been published [7]; briefly, sarcopenia was more frequent in men, and associated to chronic lung diseases, cancer, lower BMI, and previous hospital admissions, whereas frailty was more frequent in women and associated to a higher number of polypathology categories, chronic pain, anxiety, and pressure ulcers; both phenotypes shared association with age, asthenia, and lower BI scores.

### 3.1. Oxidative Stress Markers

Median catalase and GR activity were 53 nmol/min/mL (IQR = 20–83), and 9.8 U/mL (IQR = 6.6–13.2), respectively. Total antioxidant activity against ROS was 2.4 mM Trolox equivalents (IQR = 1.8–3). Finally, median SOD activity was 4.6 U/mL (IQR = 2.8–6.6).

Differences of oxidative stress markers in patients with sarcopenia, frailty, or those with both conditions with respect to those without sarcopenia, robust, or those without both conditions are detailed in Table 2.

### 3.2. Absolute Telomere Length Analysis

Mean aTL was 2 kbases per telomere (IQR = 0.1–55). Differences of aTL in patients with sarcopenia, frailty, or those with both conditions with respect to those without sarcopenia, robust, or those without both conditions are also detailed in Table 2.

### 3.3. Apoptosis

Apoptosis by means of DNA fragmentation analysis was not present in any of the patients included in the study.

### 3.4. Functional Parameters, Death Risk by PROFUND Index and Survival according to Different Molecular Parameters

A worse functional status by means of lower Barthel’s index score was associated to shorter telomere length (Beta = 1.25 (1.07–1.34)); *p* = 0.001), but not with any of the oxidative stress markers.

A higher death risk by means of PROFUND index was associated to shorter telomere length (Beta = 0.5 (0.14–0.65); *p* = 0.001) and to a higher GR activity (Beta = 1.7 (1.2–2); *p* = 0.04). On the other hand, a lower number of survival days was associated to shorter telomere length (Beta = 1.2 (1.01–1.32); *p* = 0.003) and to a higher GR activity (Beta = 0.3 (0.1–0.24)); *p* = 0.02).

## 4. Discussion

In the present study, we have detected enhanced oxidative stress and significant telomere shortening in PP with sarcopenia, frailty, or both syndromes combined. On the contrary, no evidence of apoptosis was detected.

Sarcopenia was prevalent in our cohort of polypathological patients and was associated to a significant higher SOD activity; other oxidative stress markers activity was also elevated, and the TAC-ROS decreased, but differences in these last were not significant. In the same way, we observed a significant telomere length shortening in these patients compared to other PP without sarcopenia. These results are highly concordant with the pathogenesis of sarcopenia in the elderly as already demonstrated by other authors [24,25,26,27]. Many authors have compared these markers among elderly and young people [28]; in the present study we have also detected important differences among elderly patients with chronic conditions with or without sarcopenia. These findings could have two major clinical applications: first, to use them as biological markers of sarcopenia in the elderly compared to persons of the same age; and second, to guide future treatments towards these targets in order to avoid or delay the development of sarcopenia. With respect to oxidative stress, SOD was the marker with the largest differences among PP with or without sarcopenia. As a matter of fact, SOD has been already strongly linked to muscular weakness, muscular wasting, and sarcopenia in clinical and experimental scenarios [29,30,31]; in this sense, among others, probably SOD could be the optimal oxidative stress marker in the evaluation of sarcopenia.

Frailty was also highly prevalent in the studied PP cohort and was associated to a significant increase in SOD activity and a decreased plasma TAC-ROS. It was also associated to a significant telomere length shortening compared to other PP without frailty. The deep relation between sarcopenia and frailty is already known; they share molecular and physiological pathways, symptoms, signs, and clinical phenotypes [32,33], so the presence of these molecular alterations in both of them is biologically coherent. In this case, we also observed a decreased antioxidant fitness of the plasma in frail PP. As main differences, frailty is more age related, whereas sarcopenia is also related to disease, starvation, and disuse [34]; additionally, despite criteria defining the two conditions overlap, frailty requires weight loss, whereas sarcopenia requires muscle loss [34,35].

In PP with sarcopenia and/or frailty we have observed the coexistence of telomere shortening and enhanced oxidative stress. There is accumulating evidence of the role of oxidative stress in DNA damage and telomere shortening with aging and chronic diseases [36]. These changes have been observed in humans, as well as in mouse models and cell cultures [36]. There are probably mixed mechanisms in this narrow relation of oxidative stress and telomere length. In aging and in many chronic conditions, processes associated to chronic inflammation play a nuclear role. Chronic inflammation is characterized by higher oxidative stress in affected tissues and circulating plasma. This may lead to direct cell DNA damage, including telomere regions. Additionally, inflammatory states are associated to enhanced necrosis and cellular regeneration cycles, and this increased cell turnover directly affects telomere length [37,38,39,40,41,42]. Many authors already point out that targeting oxidative stress could be of notable benefit in telomere length maintenance, especially in populations with chronic conditions like patients in the present study [39,40,41,42].

We did not detect any DNA fragmentation in our patients’ leucocytes, so no apoptosis evidence could be detected in PP’s blood samples by this technique. Apoptosis pathways have been classically associated to sarcopenia and frailty and nowadays are considered one of the main causes of these two syndromes [43,44]. As a matter of fact, there is multiple evidence of apoptosis presence in muscle tissue of experimental animal models, as well as in humans with sarcopenia [45,46,47,48,49]. Nevertheless, no information is available about the presence of apoptosis evidence in blood leukocytes of patients with sarcopenia and/or frailty. Some authors have described indirect apoptosis pathways data in blood leucocytes in elderly and in patients with dementia (like less resistance to experimental apoptosis inducers; senescence of CD8+ T-cells; and increased expression of HLA-DR, CD95, and Bcl-2 in CD3+ lymphocytes) [50]. Recently, increased ROS production and DNA fragmentation has been observed in blood monocytes of atherosclerotic mice, uprising again the interrelations of oxidative stress and apoptosis signaling [51]. Apoptosis will for sure be present in muscle tissues of patients with sarcopenia and frailty, like enhanced oxidative stress and telomere shortening. Nevertheless, according to our data, an easy detection of its presence in blood samples from these patients is probably not useful in the clinical setting, and demonstrating it in tissue specimens is not clinically justified.

A poorer functional status, higher mortality risk, and less survival days in the 12-month follow-up were associated to shorter telomere length; besides, mortality risk and survival days were also associated to enhanced GR activity. These data are in concordance with previous studies in which telomere shortening has been associated to poorer survival in cancer, diabetes, cardiovascular diseases, and even to higher all-cause mortality [52,53,54,55,56]; additionally, oxidative stress has also been related to poor health outcomes in many clinical scenarios, and to all-cause mortality [57,58,59,60]. Our data confirm this deleterious relationships with sarcopenia and frailty in patients with multimorbidity, as well as the association to poorer functional status. Some authors have already claimed the clinical usefulness of biomarkers’ panels including aTL, if we want to accurately assess and predict outcomes in vulnerable aged populations [61]. We suggest including also oxidative stress markers in these panels, mainly GR, TAC-ROS, and SOD.

This study has some limitations that should be noted. The results could be limited by the number of patients, but on the other hand, the cohort was recruited in various centers, was homogeneous, and probably represents adequately hospital-based populations with moderate–severe multimorbidity. Additionally, the studied biomarkers are also associated to some of the chronic conditions of the included patients and could raise the question of their real correlation to sarcopenia and frailty; this issue always underlies the frailty and sarcopenia phenotypes, since they have multiple concurrent causes, with a prominent role of debilitating chronic diseases; in our opinion, they behave as parts of the same clinical-molecular syndrome; as a matter of fact, the term “inflamm-aging” is already established, and probably in the future, it will be necessary to add chronic conditions and call it “inflamm-chronic-aging”.

In conclusion, oxidative stress and telomere shortening, but not apoptosis markers, were enhanced in blood samples of polypathological patients with sarcopenia and/or frailty with respect to those patients without these two geriatric syndromes. Telomere shortening was associated to functional decline, and both, oxidative stress markers and telomere shortening, were associated to higher mortality risks and decreased survival. Both of these biomarkers could be useful in the clinical evaluation of vulnerable patients prone to sarcopenia and frailty and of potential interest as therapeutic targets.

## Figures and Tables

**Table 1 jcm-09-02669-t001:** Main clinical and biological features of a multicenter sample of 444 polypathological patients recruited for sarcopenia and frailty assessment.

Clinical Features	Mean (SD)/Median -IQR-/*N* (%)
Number of defining categories (major diseases) per patient	2.5 (0.5)
Prevalence of defining categories (major diseases)	
Heart diseases	374 (84.6%)
Kidney/autoimmune diseases	202 (45.7%)
Lung diseases	183 (41.4%)
Neurological diseases	133 (30.1%)
Peripheral arterial disease/diabetes with neuropathy	80 (18.1%)
Neoplasia/chronic anemia	70 (15.8%)
Degenerative osteoarticular disease	43 (9.7%)
Liver disease	28 (6.3%)
Number of other comorbidities per patient	5.9 (2.3)
Most frequent comorbidities	
Hypertension	380 (86%)
Dyslipemia	232 (52.5%)
Diabetes with no visceral involvement	216 (49%)
Atrial fibrillation	178 (40%)
Obesity	159 (36%)
Anxiety and depressive disorders	74 (17%)
Benign prostate hyperplasia	64 (14.5%)
Frequent symptoms	
Fatigue	304 (70%)
Anorexia	212 (48%)
Insomnia	194 (44%)
Chronic pain	178 (40%)
Cough	158 (36%)
Patients with basal III-IV class of NYHA//III–IV class of mMRC	128 (29%)
PROFUND index	6 -6-
Basal Barthel’s Index	66 (30)
BMI (Kg/m^2^)	30 (6.6%)
Main biological parameters	
Hemoglobin (d/dL)	11.3 (2)
Creatinin (mg/dL)	1.26 (1)
Albumin (g/dL)	3.2 (0.9)
Cholesterol (mg/dL)	151 (42)
Triglicerydes (mg/dL)	116 -80-
Vitamin D (ng/mL)	11 -17-
Leucocytes (n°/µL)	8000 -4000-
Lymphocytes (n°/µL)	1200 -400-

SD: standard deviation; IQR: interquartile range; NYHA: New York Heart Association; mMRC: Medical Research Council. BMI: body mass index.

**Table 2 jcm-09-02669-t002:** Differences of oxidative stress markers, telomere length, and apoptosis markers in patients with multimorbidity according to their sarcopenia and frailty assessment.

Molecular Parameter	Sarcopenia and Frailty Assessment		*p*
Sarcopenia	Not Sarcopenic	Sarcopenic	
Oxidative stress marker			
CAT	52 (12.6–82) *	58 (29.4–87.8)	0.16
GR	9.8 (6.8–13.4)	9.9 (5.5–13)	0.45
TAC-ROS	2.42 (1.8–3)	2.29 (1.75–2.8)	0.12
SOD	4.4 (2.7–6.5)	5.8 (3.7–6.9)	0.02
Absolute telomere length	4.96 (0.7–19)	1.65 (0.6–3.9)	0.001
Apoptosis (WBC DNA fragmentation)	No evidence	No evidence	-
**Frailty**	**Robust**	**Frail**	
Oxidative stress marker			
CAT	55.6 (21.7–80)	51.4 (19–83.5)	0.6
GR	9.1 (5.3–9)	10.2 (6.9–13.5)	0.12
TAC-ROS	3.5 (1.6–9)	3.3 (1.9–3)	0.044
SOD	3.8 (2.3–6.2)	5.1 (3.2–7)	0.002
Absolute telomere length	5.7 (1.7–19)	1.5 (0.6–3.4)	<0.0001
Apoptosis (WBC DNA fragmentation)	No evidence	No evidence	-
**Sarcopenia and Frailty**	**Not Sarcopenic and Robust**	**Sarcopenic and Frail**	
Oxidative stress marker			
CAT	46.4 (46.5–77.5)	51.5 (26.2–79)	0.2
GR	9.7 (6.7–13.2)	10.1 (5.9–14)	0.5
TAC-ROS	2.4 (1.8–3.1)	2.2 (1.8–2.8)	0.08
SOD	4.4 (2.8–6.4)	5.7 (4.1–6.4)	0.0012
Absolute telomere length	6.5 (0.7–20)	1.5 (0.6–3.8)	<0.0001
Apoptosis (WBC DNA fragmentation)	No evidence	No evidence	-

CAT: catalase (nmol/min/mL); GR: Glutathione reductase (U/mL); TAC-ROS: total antioxidant activity against reactive oxygen species (mM Trolox equivalents); SOD: superoxide dismutase (U/mL); absolute telomere length (kbases/telomere); * Interquartile range; WBC: white blood cells; DNA: deoxyribonucleic acid.

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
