# Peer review of "Oxidative Stress, Telomere Shortening, and Apoptosis Associated to Sarcopenia and Frailty in Patients with Multimorbidity"

_jcm, 2020, doi:10.3390/jcm9082669_

Round 1

Reviewer 1 Report

Thank you for all improvements.

Reviewer 2 Report

None.

This manuscript is a resubmission of an earlier submission. The following is a list of the peer review reports and author responses from that submission.

Round 1

Reviewer 1 Report

The manuscript "Oxidative stress, telomere shortening..." describes the findings about the disturbances of some biomarkers in two relevant conditions (frailty and sarcopenia) that can converge in the same individuals. The study is done in a clinical survey composed by 444 patients attended in Internal MNedicine and Geriatrics facilities.

The topic is of interest taking into account that both entities are well-recognized modifiable risk factors for several adverse outcomes. However the manuscript have severe flaws and methodological concerns.

Major concerns

1) The diseases and conditions accompanying the presence of frailty and/or sarcopenia are linked to the markers assessed by the authors. However there is not any adjustment for these factors in the compoarison between the values in patients with/without for anyone of the biomarkers. Thus, it is not posible to know if the differences/associations are due to the presence of frailty/sarcopenia or to any of the coexisting diseases

2) Regarding risk of death, authors tell that they are going to measure it in two different ways: as a dichotomous variable and as time of survival. But in the section of statistics they only mention to the second option, as it is also the case in results. Furthermore although they use lineal regression model, again they do not adjust for confounders nor show beta values. Taking into account that they have the time for the event, they could show their findings through Hazard Ratios or, additionally Kaplan-Meier curves

3) They compare frail against robusts. What about pre-frail? Were they excluded? Or the compariuson is between frail vs non-frail (robust plus pre-frail)? If so, is there any gradation in the differences)?

4) What does mean "polypathological"? Is it the same than multimorbidity? Does this mean that the criteria is to have 2 or more diseases/conditions? If so, how is it posible that the mean of defining categories is 2.5 (0.8)?

Minor concerns

1) In the introduction authors  tell that "most sarcopenic patients are frail" (line 53). This is a controversial assertion as other authors have found the opposite in community-dwelling surveys (i.e., Davies et al., JAMDA 2018)

2) When authors support the the relationship between oxidative stress and frailty/sarcopenia the three references they mention are review papers. It would be worthy to use papers communicating original observations as references (i.e., Kameda et al., PNAS 2020)

3) Mean age of the population is 77.3. Taking into account that part of the population was recruited in geriatric facilities, where PROFUND does not seem to perform appropriately (Diez-Manglano et al., 2016), why do you choose this tool for assessing the risk in this particular population?

4)  Sample size is 444. But in the first paragraph of the results section the sum of the three groups is 445 and the total of the percentages is 102%. Please, review

5) In your sample, 80% of the population are frail and 40% show sarcopenia, figures largely exceeding the usual ones in the communityt. Do you think that this fact can be a source of bias? 

Author Response

REVIEWER 1:

The manuscript "Oxidative stress, telomere shortening..." describes the findings about the disturbances of some biomarkers in two relevant conditions (frailty and sarcopenia) that can converge in the same individuals. The study is done in a clinical survey composed by 444 patients attended in Internal MNedicine and Geriatrics facilities.

The topic is of interest taking into account that both entities are well-recognized modifiable risk factors for several adverse outcomes. However the manuscript have severe flaws and methodological concerns.

ANSWER: Thank you for your kind comments about our work.

Major concerns

1) The diseases and conditions accompanying the presence of frailty and/or sarcopenia are linked to the markers assessed by the authors. However there is not any adjustment for these factors in the compoarison between the values in patients with/without for anyone of the biomarkers. Thus, it is not posible to know if the differences/associations are due to the presence of frailty/sarcopenia or to any of the coexisting diseases

We generically agree with this reviewer's doubt. The studied biomarkers are also associated to some of the chronic conditions of the included patients and could raise the question of their real correlation to sarcopenia and frailty.

This issue always underlies in the frailty and sarcopenia phenotypes, since they have multiple concurrent causes, with a prominent role of debilitating chronic diseases; in our opinion, they behave as parts of the same clinical-molecular syndrome; as a matter of fact, the term 'inflamm-aging' is already established and probably in the future it will be necessary to add chronic conditions and call it 'inflamm-chronic-aging'.

We think, that trying to separate if molecular biomarkers statistically correlate independently to chronic debilitating conditions, or to sarcopenia or frailty could introduce us in a conceptual error, because probably they are links of the same chain, that are being explored from different angles as other author have already mentioned (Cesari M, et al. Front Aging Neurosci 2014).

Nevertheless, and according to this reviewer's suggestion we have incorporated this issue in the Discussion Section, y Limitations of the study.

2) Regarding risk of death, authors tell that they are going to measure it in two different ways: as a dichotomous variable and as time of survival. But in the section of statistics they only mention to the second option, as it is also the case in results. Furthermore although they use lineal regression model, again they do not adjust for confounders nor show beta values. Taking into account that they have the time for the event, they could show their findings through Hazard Ratios or, additionally Kaplan-Meier curves.

According to this reviewer's suggestion we have eliminated the mortality analysis as dichotomous variable in the methods section, and have incorporated de beta values of the lineal regression model.

In order to plot K-M curves a categorization in tertiles or quartiles of all oxidative stress and telomeres length measures are to be done. There is a difficulty in establishing the cut-off points; in addition, for detailing the K-M curves we should add 5 new figures to the manuscript (catalase, SOD, GR, TAC-ROS, and telomere length). This would enlarge the article, and difficult its reading. Thus, we have decided not to include K-M curves.

3) They compare frail against robusts. What about pre-frail? Were they excluded? Or the comparison is between frail vs non-frail (robust plus pre-frail)? If so, is there any gradation in the differences)?

Yes, the comparisons are between frail patients versus non-frail (which contained robust patients plus pre-frail patients). When comparing among the three subgroups a non-significant gradation tendency can be observed, probably due to the less number of patients in the pre-frail and robust subcategories.

4) What does mean "polypathological"? Is it the same than multimorbidity? Does this mean that the criteria is to have 2 or more diseases/conditions? If so, how is it posible that the mean of defining categories is 2.5 (0.8)?

Yes. Definition of polypathological patient comprises all patients who suffer from two o more chronic diseases included in a list of 8 the following categories:

Functional definition of Polypathological Patient: the patient who suffers chronic diseases included in two or more of the following clinical categories.

CATEGORY A

A.1 Chronic heart failure with past/present stage II dyspnea of NYHA1.

A.2 Coronary heart disease

CATEGORY B

B.1 Vasculitides and/or systemic autoimmune diseases

B.2 Chronic renal disease (creatininaemia >1.4/1.3 mg/dl in men/women or proteinuria2, during ≥3 months

CATEGORY C

Chronic lung disease with past/present stage 2 dyspnea of MRC3, or FEV1<65%, or basal SatO2 ≤ 90%

CATEGORY D

D.1 Chronic inflammatory bowel disease

D.2 Chronic liver disease with evidence of portal hypertension4

CATEGORY E

E.1 Stroke

E.2 Neurological disease with permanent motor deficit, leading to severe impairment of basic activities of daily living (Barthel’s index <60).

E.3 Neurological disease with permanent moderate-severe cognitive impairment (Pfeiffer’s test with ≥5 errors).

CATEGORY F:

F.1 Symptomatic peripheral artery disease

F.2 Diabetes mellitus with proliferate retinopathy or symptomatic neuropathy

CATEGORY G:

G.1 Chronic anemia (Hb< 10g/dL during ≥3 months) due to digestive-tract losses or acquired haemopathy not tributary of treatment with curative intention.

G.2 Solid-organ or Hematological active neoplasia not tributary of treatment with curative intention.

CATEGORY H:

Chronic osteoarticular disease, leading to severe impairment of basic activities of daily living (Barthel’s index <60)

Slight limitation of physical activity. Comfortable at rest, but ordinary physical activity results in fatigue, palpitation, or dyspnea.

2 Albumin/Creatinine index > 300 mg/g, microalbuminuria >3mg/dl in urine, albumin >300 mg/day in 24-h urine, or albuminuria/min >200 microg/min.

3 Short of breath when hurrying or walking up a slight hill

4 Presence of clinical, analytical, echographic, or endoscopic data of portal hypertension.

In the area of chronic conditions, polypathological patients (PP) are nowadays the a clinical paradigm of the emergence of chronic conditions in our societies. They acomplish all criteria of the so called 'populations with complex chronic diseases' because of their high prevalence in most clinical arenas, their advanced age (mean age of PP in multicenter cohorts rounded 75-80 years), their complexity, disease and symptom burden, clinical vulnerability, poor health-related quality of life, tendency towards functional deterioration, and high mortality rates. So polypathology is not the same as multimorbidity, because it adds the clinical value of 'complexity and severity' to the simple 'sum-up' concept of multimorbidity.

All polypathological patients have multimorbidity, but not all patients with multimorbidity are polypathological patients. For instance a patient with hypertension and hypothyroidism can be considered having multimorbidity, but he is not a polypathological patient. On the other hand a patient with chronic heart failure with functional class II-III and chronic kidney disease is a polypathlogical patients and obviously has also multimorbidity.

Sorry for the typographical mistake in the standard deviation. We have amended it.

Minor concerns

1) In the introduction authors  tell that "most sarcopenic patients are frail" (line 53). This is a controversial assertion as other authors have found the opposite in community-dwelling surveys (i.e., Davies et al., JAMDA 2018)

Yes, this was an affirmation picturing the results of the clinical study (ref number #7). Nevertheless, according to this reviewer's suggestion we have incorporated this controversy and reference in the revised version of the manuscript.

2) When authors support the the relationship between oxidative stress and frailty/sarcopenia the three references they mention are review papers. It would be worthy to use papers communicating original observations as references (i.e., Kameda et al., PNAS 2020)

According to this reviewer's suggestion, we have incorporated this reference with the number #26.

3) Mean age of the population is 77.3. Taking into account that part of the population was recruited in geriatric facilities, where PROFUND does not seem to perform appropriately (Diez-Manglano et al., 2016), why do you choose this tool for assessing the risk in this particular population?

With respect to this issue, most patients were recruited in Internal Medicine areas, and conformed a hospital-based cohort. All included patients accomplished the definition of polypathological patient, so the PROFUND index was the best choice to predict their death-risk.

PROFUND index was originally developed and validated to predict 12-month mortality in hospital-based patients with polypathology, but its generalizability has been established later by means of historical, geographic, spectrum, and follow-up transportability (in primary care polypathological patients, in other geographical areas, in patients with heart failure, and in shorter (predicting hospitalization episode mortality) and longer (four years) periods of follow-up). Recently a systematic revision of prognostic indices in patients with multimorbidity catalogued it as of satisfactory quality among other 35 evaluated tools.

- P. Bohorquez Colombo; et al. Validation of a prognostic model for polypathological patients (PP) in Primary Health Care: PROFUND STUDY-AP. Atención Primaria 2014; 46 (Suppl 3): 41-48.

- Bernabeu-Wittel M, et al. Validation of PROFUND prognostic index over a four-year follow-up period. Eur J Intern Med. 2016; ;36:20-24.

- Díez-Manglano J, et al. External validation of the PROFUND index in polypathological patients from internal medicine and acute geriatrics departments in Aragón. Intern Emerg Med 2015; 10:915-26.

- López-Garrido MA, et al. Prevalence of comorbidities and the prognostic value of the PROFUND index in a hospital cardiology unit. Rev Clin Esp 2017; 217:87-94.

- Martín-Escalante MD, et al. Validation of the PROFUND index to predict early post-hospital discharge mortality. QJM 2019; 112:854-60.

- Stirland LE, et al. Measuring multimorbidity beyond counting diseases: systematic review of community and population studies and guide to index choice. BMJ 2020;368:m127. http://dx.doi.org/10.1136/bmj.m127.

4)  Sample size is 444. But in the first paragraph of the results section the sum of the three groups is 445 and the total of the percentages is 102%. Please, review.

Sorry for the misunderstanding. Of course these three groups do not have to sum up the total N of the included patients, it would be really disturbing. As a matter of fact, some of the sarcopenic patients were also frail, and vice versa, and this is what the data intend to express.

Out of the 444 patients, 278 (62.6%) were frail, 97 (19.8%) were sarcopenic, and the remaining 69 (15.6%) were robust.

Additionally 80 (18% of the whole cohort) of the patients with sarcopenia or frailty, had simultaneously both phenotypes.

In order to clarify this fact for readers, and following reviewer's suggestion we have reorganized this information in the revised version of the manuscript.

5) In your sample, 80% of the population are frail and 40% show sarcopenia, figures largely exceeding the usual ones in the communityt. Do you think that this fact can be a source of bias?

The explanation to these sarcopenia and frailty rates lies in the population included in the study. This was a hospital-based population (in these populations both phenotypes are more prevalent than in community-dwelling elders), with polypathology (chronic conditions and their severity play also a substantial role in the prevalence of these syndromes), and the recruitment (a vast majority of patients were recruited at hospital discharge).

We think, that this is not a source of bias, but on the contrary it enriches our view of sarcopenia and frailty, because it reflects an epidemiological fact and a scope to these prevalent syndromes in hospital environments.

Reviewer 2 Report

General:

The authors investigated a highly relevant and important issue, namely the determination of biologically accessible markers and their association with social burden-related phenotypes, e.g. sarcopenia and frailty. The study is well designed and –performed and adds novel (clinically) relevant information to the identification and further characterization of sarcopenia and frailty.

Specific:

Abstract: The study population seems to be heterogeneous in terms of percentage related to the focused disease phenotypes. Could the authors add some more information about the study population (not in the abstract, but in the methods section) in order to clarify that the percentages related to the respective phenotypes influence the reported results.

Introduction: lines 47-53 – It would be beneficial for the reader, if the authors highlight commonalities as well as differences between the two focused disease phenotype.

Methods: lines 130-135 – Does this SOD kit really determine the active SOD or the protein level itself without any explanatory power about its activity?

Results: Table 1 – Could the authors include a discussion section that highlights the influence of the observed comorbidities on the measured biological parameters? That could be beneficial.

Author Response

REVIEWER 2:

General:

The authors investigated a highly relevant and important issue, namely the determination of biologically accessible markers and their association with social burden-related phenotypes, e.g. sarcopenia and frailty. The study is well designed and –performed and adds novel (clinically) relevant information to the identification and further characterization of sarcopenia and frailty.

 ANSWER: Thank you for your kind comments about our work.

Specific:

Abstract: The study population seems to be heterogeneous in terms of percentage related to the focused disease phenotypes. Could the authors add some more information about the study population (not in the abstract, but in the methods section) in order to clarify that the percentages related to the respective phenotypes influence the reported results.

This manuscript details the molecular sub-study of a multicenter cohort. All clinical and epidemiological deails are already specified previously in: (7). Bernabeu-Wittel M, González-Molina Á, Fernández-Ojeda R, Díez-Manglano J, Salgado F, Soto-Martín M, Muniesa M, Ollero-Baturone M, Gómez-Salgado J. Impact of Sarcopenia and Frailty in a Multicenter Cohort of Polypathological Patients. J Clin Med 2019; 8: 535-547.

Nevertheless, and according to this reviewer's suggestion we have resumed the essentials and referred to this already published information in the methods and results section of the revised version of the manuscript.

Introduction: lines 47-53 – It would be beneficial for the reader, if the authors highlight commonalities as well as differences between the two focused disease phenotype.

According to this reviewer's suggestion we have highlighted commonalities as well as differences between frailty and sarcopenia in the Introduction section of the revised version of the manuscript.

Methods: lines 130-135 – Does this SOD kit really determine the active SOD or the protein level itself without any explanatory power about its activity?

 Yes, this kit determines the SOD activity and measures all types of SOD, by detecting the dismutation of superoxide radicals (generated by xantin oxidase).

This specifications are detailed in the Methods Section:

'Superoxide dismutase activity (U/mL) was measured in patients' plasma using the colorimetrical absorbance procedure provided by Cayman's Superoxide Dismutase Assay Kit, Item No.706002 (Cayman Chemical, Ann Arbor, Michigan USA). The method utilizes a tetrazolium salt for detection of superoxide radicals generated by xanthine oxidase and hypoxanthine. One unit of SOD is defined as the amount of enzyme needed to exhibit 50% dismutation of the superoxide radical. This assay measures all types of SOD (Cu/Zn-SOD, Mn-SOD, and Fe-SOD).'

Results: Table 1 – Could the authors include a discussion section that highlights the influence of the observed comorbidities on the measured biological parameters? That could be beneficial.

According to this reviewer's suggestion, we have included this important aspect in the discussion section of the revised version of the manuscript, in the limitations of the study.

Reviewer 3 Report

Abstract

-Please correct the "(" between line 21 and 23.

- "Both oxidative stress markers (GR) and telomere shortening were associated" please, add the P value

Introduction

- please add the study hypotheses

- despite not being the objective of the study maybe the authors should explore that is the most cases, the performing physical exercise reduce sarcopenic and others diseases

PATIENTS AND METHODS

- Who collected the data and perform the sarcopenia evaluations? Please add some information.

- All patients were followed during a 12 month period by who and how?

Discussion is well described, please add the limitations of the study.

Author Response

REVIEWER 3:  ANSWER: Thank you for your kind comments about our work.

Abstract

-Please correct the "(" between line 21 and 23.

- "Both oxidative stress markers (GR) and telomere shortening were associated" please, add the P value

According to these reviewer's suggestion we have corrected the "(" and included P values in the abstract.

Introduction

- please add the study hypotheses

- despite not being the objective of the study maybe the authors should explore that is the most cases, the performing physical exercise reduce sarcopenic and others diseases

 According to these reviewer's suggestion we have added the study hypotheses, and also included a sentence referring to the beneficial effects of physical activity on sarcopenia and frailty.

PATIENTS AND METHODS

- Who collected the data and perform the sarcopenia evaluations? Please add some information.

- All patients were followed during a 12 month period by who and how?

 This manuscript details the molecular sub-study of a multicenter cohort study previously reported, in which all clinical details were specified. 7. Bernabeu-Wittel M, González-Molina Á, Fernández-Ojeda R, Díez-Manglano J, Salgado F, Soto-Martín M, Muniesa M, Ollero-Baturone M, Gómez-Salgado J. Impact of Sarcopenia and Frailty in a Multicenter Cohort of Polypathological Patients. J Clin Med 2019; 8: 535-547.

Nevertheless, and according to these reviewer's suggestion we have added this information in the revised version of the manuscript.

Discussion is well described, please add the limitations of the study.

According to this reviewer's suggestion we have added the limitations of the study in he Discussion section of te revised manuscript.